# Reentrant melting of scarred odd crystals by self-shear

Uttam Tiwari [1] ✉, Pragya Arora [1], A. K. Sood [2,3], Sriram Ramaswamy [2,4], Rituparno Mandal[5] & Rajesh Ganapathy [1,3,6] ✉

Spatial confinement can induce geometrical frustration in condensed phases, giving rise to topological defects that confer materials with new and exotic properties. Here, we experimentally uncover the effect of confinement-induced defect strings termed 'grain boundary scars' on the behavior of dense two-dimensional assemblies of granular spinners, a canonical odd elastic solid. The spatial arrangement of these scars fundamentally reshapes the flows triggered by chiral activity. Specifically, they cause the topologically protected edge flows - a ubiquitous feature of confined spinner assemblies - to decouple from the bulk. Strikingly, increasing the net chiral activity of the system by tuning the ratio of counterclockwise to clockwise spinners caused spontaneous self-shearing. The resulting odd radial stresses led to a chiral activity-mediated reentrant melting transition at a fixed areal spinner density. Our findings open new avenues for exploiting geometrical frustration to elicit novel responses from odd elastic solids.

Topological defects are often the orchestrators of material behavior[1]. While their established role lies in tuning the physical properties of passive materials[2,3], there is growing interest in extending this paradigm to active matter systems[4,5]. A simple route to create these defects is through geometrical frustration[6], achieved by confining the system within a carefully chosen boundary shape that prevents the preferred local order of the system from tiling space. These defects persist in the system even in its ground state - a consequence of Euler's theorem[7]. A classic example is the appearance of twelve pentagonal disclination defects in a triangular lattice on a spherical surface, as seen in viral capsids[8], colloidosomes[9–11], and the familiar soccer ball. Six such defects emerge on confining the same lattice within a two-dimensional (2D) circular boundary[12,13]. To relieve the elastic strain energy on exceeding a threshold system size, these disclinations bind to dislocations, forming 1D defect strings called 'grain boundary scars'[9–11,14,15], which in turn govern the yielding behavior of these frustrated crystals[16]. In the realm of active matter, geometrical frustration effects are more dramatic. In these out-of-equilibrium systems, particularly in

those made of elongated building blocks, defects themselves can be motile entities[17–21]. This has spurred a surge of research exploring how boundary geometry can be harnessed to create and manipulate these defects, ultimately enabling global control over active flows and turbulence[22–35].

Inspired by these studies, here we examined how confinement-induced topological defects alter the collective behavior of 2D dense assemblies of active spinners. By breaking both parity and time-reversal symmetries, this system allows for the existence of novel odd material moduli that are forbidden in equilibrium[36,37]. These odd moduli generate transverse couplings between mechanical perturbations and responses; for example, compressing the assembly can induce a net torque density, driving spontaneous rotation. This behavior originates from transverse forces generated by friction between the spinning elements[38,39]. While spinners offer a specific realization of oddness, odd mechanics is a more general consequence of chirality and broken time-reversal[40,41]. The presence of odd moduli results in exotic new phenomena: topologically protected

¹Chemistry and Physics of Materials Unit, Jawaharlal Nehru Centre for Advanced Scientific Research, Jakkur, Bangalore, India. ²Department of Physics, Indian Institute of Science, Bangalore, India. ³International Centre for Materials Science, Jawaharlal Nehru Centre for Advanced Scientific Research, Jakkur, Bangalore, India. ⁴International Centre for Theoretical Sciences, Bangalore, India. ⁵Soft Condensed Matter Group, Raman Research Institute, Bangalore, India. ⁶School of Advanced Materials (SAMat), Jawaharlal Nehru Centre for Advanced Scientific Research, Jakkur, Bangalore, India. ✉e-mail: tiwari@jncasr.ac.in; rajeshg@jncasr.ac.in

unidirectional edge currents capable of transporting cargo without scattering[42], self-sustained chiral waves[43], spontaneous rotation and self-kneading of spinner crystals into whorls[44], chiral excitation and tilting of force chains[45], and emergence of a void phase[46]. Although many previous studies have investigated confined spinner packings[47–49], the fundamental role of defects arising from geometrical frustration on their emergent dynamics remains unaddressed, highlighting a key gap in our understanding.

## Results

### Tuning the oddness of dense granular spinner assemblies

To isolate how confinement-induced topological defects affect spinner ensembles, our experimental system must meet two key criteria: it must be able to host a defect-free ground state without confinement, and its odd mechanical response must be tunable. Granular spinners driven by vertical vibration suit our purpose[50,51] (Fig. 1A, see "Methods" section). Our spinners are spherical domes of diameter $a = 4.2$ mm standing on tilted legs that transduce vertical vibrations into torque and make them spin. The handedness of rotation, i.e., clockwise ($\otimes$) or counterclockwise ($\odot$), is set by the handedness of the leg tilt. Pairwise spinner interactions comprise a longitudinal hard-core repulsion and a transverse force due to contact friction. The horizontal cross-section

of the spinners is circular, and they can readily pack into defect-free triangular lattices. We induced geometrical frustration in the system by confining the spinners within a circular arena of diameter $D = 30$ cm. Our experiments spanned area fractions $0.68 \leq \phi \leq 0.79$, with the lower limit corresponding to the density beyond which the spinners began to form crystalline domains with triangular symmetry. The strength of the odd response scales with the internal torque density[36,38,44], which we tuned by adjusting the net chiral activity, $\chi$, of the spinner assembly. Transitioning from a racemic mixture (a 50:50 combination of $\otimes$ and $\odot$ spinners) to a homochiral state (all $\otimes$ or all $\odot$) tunes the internal torque density from zero to the maximum achievable while keeping all other parameters fixed. Here $\chi = \frac{N^\otimes - N^\odot}{N}$ with $N^\otimes$ and $N^\odot$ being the number of $\otimes$ and $\odot$ spinners, respectively, and $N = N^\otimes + N^\odot$. Fig. 1A shows a snapshot of the system for $\phi = 0.72$ and $\chi = 0$. For $0 \leq \chi < 1$, the spinners remained well-mixed and did not show any tendency to phase separate (Supplementary Fig. 1). At the largest area fraction studied ($\phi = 0.79$), the system contains $N = 4000$ spinners, making it an order of magnitude larger than previous granular experiments[48–51].

We complement the experiments with discrete element method (DEM) simulations that capture key features of our granular assemblies. In our minimal model, the spinners are monodisperse disks

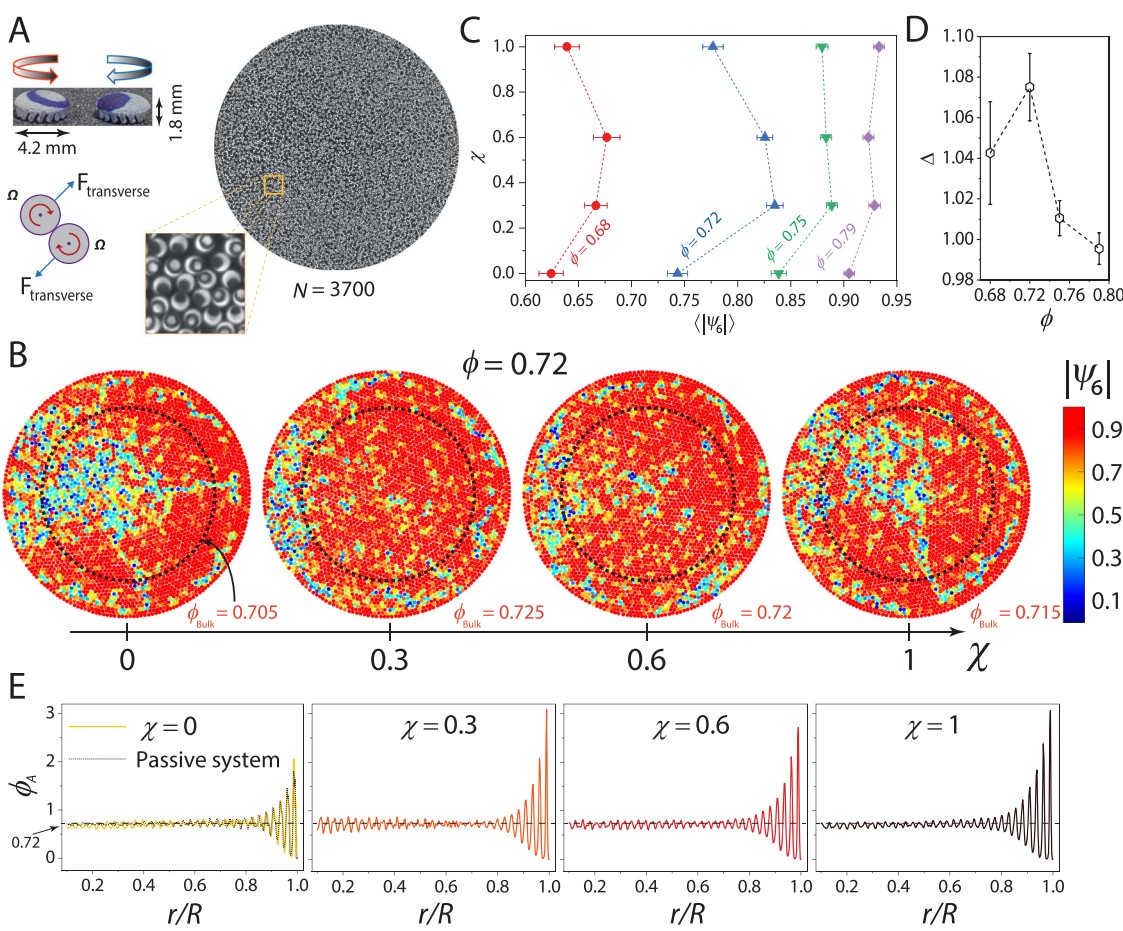

**Fig. 1 | Chiral activity drives reentrant melting of spinner crystals. A** Top left panel: Picture of 3D-printed clockwise ($\otimes$) and counterclockwise ($\odot$) rotating spinner; arrows show the spin direction. Right panel: Representative image of a dense packing of spinners, interacting via transverse forces, for net chiral activity, $\chi = 0$, and area fraction, $\phi = 0.72$. The zoomed-in view shows circles and dots marked on the $\odot$ and $\otimes$ spinners, respectively, to track their rotation. **B** The panels show spinners colored as per the magnitude of the hexagonal bond-order parameter, $\langle |\psi_6| \rangle$, for $\phi = 0.72$ at different $\chi$ values. The dashed circle delineates the bulk from the boundary and has a cutoff radius, $r_c = 0.7R$, where $R$ is the system's radius.

This cutoff corresponds to an $r$ where layering due to confinement is negligible (Fig. 3C). The averaged areal spinner density of the bulk, $\phi_{Bulk}$, is also indicated. **C** Shows the variation in $\langle |\psi_6| \rangle$ on increasing $\chi$ for different $\phi$ values. Here, $\langle \rangle$ denotes an average over all the spinners in the bulk and at all times. The error bars denote the standard error. **D** Strength of reentrance, $\Delta$ (shown in **C**), versus $\phi$. **E** Shows the annular radial density profile, $\phi_A(r)$, with $r/R$, for different $\chi$ values at $\phi = 0.72$. $\phi_A(r)$ for a packing of passive disks at the same $\phi$ is also shown. Here, $r$ is the outer radius of the annulus, with width $0.09d_p$, where $d_p$ is the particle diameter.

within a circular confinement ("Methods" section). Each spinner is subject to a torque that enforces a constant angular velocity. The equations of motion for the position and the angular coordinate of the spinners follow underdamped dynamics. Spinners interact via a repulsive normal force and a tangential friction force, computed using the spring-dashpot model.

### A chiral activity-mediated reentrant melting transition

Figure 1B captures our main observation. The panels show spinner packings with increasing $\chi$ for $\phi = 0.72$, with the colors representing the magnitude of the local hexagonal bond-order parameter, $|\psi_6^i|$. Here, $\psi_6^i = \frac{1}{N_i}\sum_{j=1}^{N_i} e^{6i\theta_{ij}(t)}$ where $N_i$ is the coordination of spinner $i$ and $\theta_{ij}$ is the angle made by the line joining the centers of spinner $i$ and its nearest-neighbor $j$ with respect to an arbitrary reference axis. What is striking is the emergence of a near-perfect *single crystal* in the bulk (region enclosed by the dashed circle) for intermediate values of $\chi$, while for the extreme values, a large pocket of liquid is evident. Tuning $\chi$ causes reentrant melting of the bulk, the imprint of which is also present in the pair-correlation function, $g(r)$ (Supplementary Fig. 2).

We recall that for equilibrium hard disks in 2D, the liquid phase is found at densities $\phi_{Eq}^{Liq} \leq 0.716$, and the crystalline phase is stable for $\phi_{Eq}^{Xtal} \geq 0.72$; the intervening phase is a hexatic[52]. While drawing direct comparisons with equilibrium systems is difficult because our system is driven, it is notable that the bulk areal density, $\phi_{Bulk} < \phi_{Eq}^{Liq}$ for $\chi \in [0, 1]$ and $\phi_{Bulk} \geq \phi_{Eq}^{Xtal}$ for $\chi \in [0.3, 0.6]$ (Fig. 1B). The system at $\phi = 0.68$ also showed a reentrant behavior. At higher densities, increasing $\chi$ enhanced crystallinity, but the system did not remelt (Fig. 1C, see Supplementary Figs. 3–5 for $|\psi_6^i|$ at other $\phi$ and $\chi$ values). In fact, the strength of this reentrance, $\Delta = \frac{\langle|\psi_6|\rangle_{\chi=0.3}}{\langle|\psi_6|\rangle_{\chi=1}}$, appears to be non-monotonic with $\phi$, with an apparent maximum near $\phi \approx 0.72$ (Fig. 1D).

### Chiral activity causes spontaneous self-shearing

Since reentrant melting of the bulk is driven by chiral activity, the underlying physics may be due entirely to odd effects, which are known to couple azimuthal flows to radial density changes[36,38,39,44]. Indeed, the radial density, $\phi_A(r)$, showed marked changes with $\chi$ (see Fig. 1E) for $\phi = 0.72$. For $\chi = 0$ (net zero torque density), $\phi_A(r)$ is nearly identical to that of a passive disk packing at the same $\phi$. The enhancement in density near the boundary is simply due to layering[53,54]. As $\chi$ increases, layering became more pronounced, and $\phi_A(r)$ in the bulk ($r/R < 0.7$) also showed clear changes. Both of which are clear signatures of additional radial stresses at play.

These observations present a natural segue into quantifying the flows that may have induced these density changes. In Fig. 2A, we show the annular angular velocity, $\omega(r)$, for different values of $\chi$ for $\phi = 0.72$. For all $\chi > 0$, we observed edge flows - a ubiquitous feature of confined spinner assemblies. These flows are a consequence of the unbalanced torques on spinners at the edge[47,48]. Additionally, the edge flows have the same handedness as the spin of the majority spinner species ($\otimes$ spinners), indicating that inter-spinner frictional interactions dominate over spinner-boundary ones[48] (see Supplementary Fig. 6). The transverse forces exerted by spinners in the penultimate layer on those in the outermost one drive the edge flow. For $\chi = 0$, these forces cancel since $N^\otimes \approx N^\odot$ in each annulus, and $\omega(r)$ fluctuated around zero for all $r/R$.

Two features in the measured $\omega(r)$ stood out. First, the magnitude of the edge flow for $\chi > 0$ dropped precipitously at $r/R \approx 0.9$ for $0.68 \leq \phi \leq 0.75$. This behavior should be contrasted with previous observations on spinner liquids and solids. In spinner liquids, the edge flow decays exponentially into the bulk since the system lacks a shear modulus, while the finite modulus of solids causes these edge flows to couple with the bulk, resulting in rigid body rotation[47]. Second, for $\phi = 0.68$ and 0.72, the system spontaneously *self-shears* (also see Supplementary Figs. 7 and 8). Specifically, for $\phi = 0.72$, while $\omega(r)$

hovered around zero for $r/R < 0.9$ at $\chi = 0.3$, it became negative and nearly constant in bulk for $\chi = 0.6$ and 1, with the magnitude of counterclockwise rotating bulk flow larger at $\chi = 1$ (see Supplementary Videos 1–4).

At higher densities ($\phi = 0.75$ and $\phi = 0.79$), the system showed neither reentrant melting nor self-shearing (Supplementary Figs. 7 and 8). For $\phi = 0.75$ and all $\chi > 0$, the magnitude of the edge current was smaller than that at $\phi = 0.72$ due to stronger inter-spinner interactions, and $\omega(r)$ decayed to zero in the bulk, while for $\phi = 0.79$, we observed rigid body rotation for $\chi > 0$. These results demonstrate that reentrant melting and self-shearing are concomitant phenomena; below, we explain their common origin.

### Grain boundary scars aid in decoupling edge and bulk flows

We first looked for cues in the structure of the spinner packings. In Fig. 2B, we show the Voronoi tessellation of the spinner packing for $\phi = 0.72$ and $\chi = 1$. The yellow hexagons represent spinners with six-fold coordination (crystalline particles), while polygons of other colors represent under- and over-coordinated spinners (disclination defects). The large defect clusters in the bulk correspond to the liquid pocket shown in Fig. 1B. Notably, at $r/R \approx 0.9$, where $\omega(r)$ fell sharply (gray annulus in Fig. 2B), we observed azimuthally aligned grain boundary (GB) scars, which also appear as strings of low $|\psi_6|$ particles in Fig. 1B (see Supplementary Fig. 9 for other system sizes). Each scar comprised an excess 5-coordinated disclination decorated by a string of dislocations. The dislocations - neutral disclination dipoles - are not a topological requirement but are present to help screen the strain field of the disclination[7,14]. We could easily identify six GB scars at intermediate chiral activity values, where crystalline order was prominent, while for extreme values ($\chi = 0$ & 1), some of the scars were part of larger defect clusters, making their identification challenging (Supplementary Fig. 10).

The GB scar configuration is determined by geometry and not activity. Crucially, we observed that vibrated passive disk packings at the same $\phi$ showed similar scar arrangements, confirming that it is the circular confinement that decides these configurations. Further, as observed in passive particle packings on a sphere, the number of excess dislocations per scar, $N_D$, increased with the system size ($D/a$) for both spinner and passive disk packings (Supplementary Figs. 9 and 11)[9,14]. Notably, our simulations of confined spinners accurately reproduced our experimental results (Fig. 2C, Supplementary Fig. 12). Importantly, even in systems where the GB scars were not well-defined due to excessive disorder ($\phi = 0.68$), the drop in $\omega(r)$ occurred in the annuli containing a higher-than-average defect density (Supplementary Figs. 13 and 14).

GB scars/defects clearly impact the coupling between the edge flow and the bulk. To see why, we note that within a continuum description of a chiral active material with strains $\partial_l u_k$ and strain-rates $\partial_l \dot{u}_k$, the stress tensor $\sigma_{ij} = -P\delta_{ij} + K_{ijkl}\partial_l u_k + \eta_{ijkl}\partial_l \dot{u}_k + \sigma_{ij}^{spin}$ (Supplementary Fig. 15)[36,44,55]. Here, $P$ is the pressure, $\delta_{ij}$ is the Kronecker delta, and $K_{ijkl}$ and $\eta_{ijkl}$ are the elasticity and the viscosity tensors, respectively, and contain all moduli consistent with broken parity and time-reversal. The last term is the antisymmetric stress due to the imposed drive torque and is given by $\sigma_{ij}^{spin} = 2\eta_R \epsilon_{ij}\Omega$, where $\eta_R$ is the rotational viscosity, $\Omega$ is the spin angular velocity field of the particles, and $\epsilon_{ij}$ is the Levi-Civita symbol[37]. A mismatch between $\Omega$ and the 2D vorticity, $(\nabla\times\mathbf{v})_z = 2\omega(r)$, where $\mathbf{v}$ is the velocity field, results in a frictional stress $\sigma_{ij}^{fric} = \eta_R \epsilon_{ij}(2\Omega - \omega)$[37]. Since $\eta_R$ results from frictional collisions between the spinners, it is proportional to the local spinner density and is expressed as $\eta_R \propto \phi^2 g(r_s)$[48,56], where $g(r_s)$ is the pair-correlation function at contact.

Here, in addition to layering, GB scars also modify the spinner density in the annuli, but only near $r/R \approx 0.9$. Figure 2D shows that $\phi_A(r/R \approx 0.9)$ in the sector harboring the scar is nearly 25% smaller than the one without it for $\phi = 0.72$ (see Supplementary Fig. 16 for $\phi = 0.68$

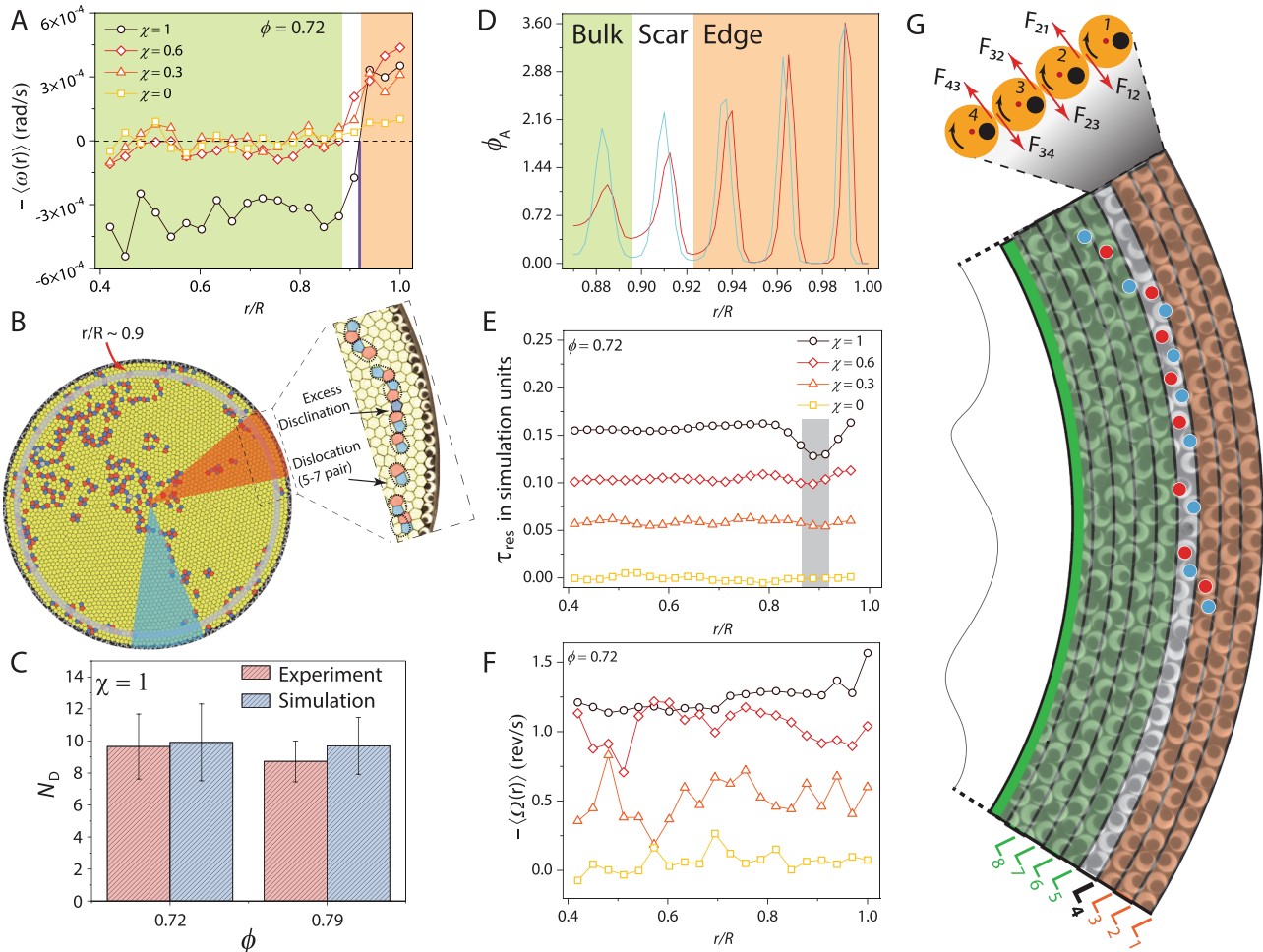

**Fig. 2 | Grain boundary (GB) scars cause spinner crystals to self-shear.** All subfigures are for $\phi = 0.72$. **A** Average angular velocity, $\omega(r)$, of spinners in annuli defined with respect to the system center versus $r/R$ for different values of $\chi$. There is a sharp drop in $\omega(r)$ at $r/R \approx 0.9$ (black vertical line) for $\chi > 0$. The bulk and the edge rotate in opposite directions for $\chi = 0.6$ & 1. The orange- and green-shaded regions correspond to the edge and bulk flow, respectively. **B** Voronoi tessellation of the spinner assembly at $\chi = 1$. The colors represent the spinner coordination number. The drop in $\omega(r)$ occurs across the annulus harboring GB scars (gray-shaded annulus). The zoomed-in view shows the composition of a GB scar. Unlike conventional grain boundaries, which terminate at the system edge, these scars terminate within the system[9]. **C** The average number of dislocations per scar, $N_D$,

for the experiment (red) and simulation (blue) at $\chi = 1$ for $\phi = 0.72$ and $\phi = 0.79$. The error bars represent the standard error. **D** Shows $\phi_A(r)$ close to the edge for $\chi = 1$ in sectors with and without a GB scar, corresponding to the red- and blue-shaded regions in (**B**), respectively. **E** Resistive torque, $\tau_{res}$, obtained from simulations for different values of $\chi$ at $\phi = 0.72$. There is a clear minimum in $\tau_{res}$ at $r/R \approx 0.9$ (gray-shaded region). **F** Shows the annular spinner spin velocity $\langle \Omega(r) \rangle$ versus $r/R$ for different $\chi$ values. **G** A snapshot of spinners near the confining boundary at $\chi = 1$. Blue and red dots are scattered over the five- and seven-coordinated spinners, respectively, to highlight the scar, which is located in layer $L_4$. The direction of the force experienced (exerted) by the annulus from (on) its neighboring one is also shown. For instance, $F_{12}$ is the force on layer $L_1$ due to $L_2$.

and $\phi = 0.75$). Consequently, the average spinner density, and hence $\eta_R$ and $\sigma_{ij}^{fric}$, are smaller in the scarred annuli compared to a system without the scars. A proxy for $\sigma_{ij}^{fric}$ that can be easily measured in our simulations is $\tau_{res}(r) = \sum_{i \neq j} \tau_{ij}$ - the resistive torque experienced by an annulus from annuli straddling it. Here, $i$ and $j$ denote particles in the annulus under consideration and in those adjacent, respectively. Fig. 2E shows $\tau_{res}(r)$ for the same $\chi$ values investigated in the experiments for $\phi = 0.72$. A pronounced minimum appears in $\tau_{res(r)}$ at $r/R \approx 0.9$ for $\chi = 1$ (see Supplementary Fig. 17 for other $\phi$ values), indicating that GB scars weaken the coupling between the edge flow and the bulk. On lowering $\chi$, both the magnitude of $\tau_{res}(r)$ and the depth of the minimum become smaller and vanish at $\chi = 0$. This trend arises because increasing the fraction of $\odot$ spinners at the expense of $\otimes$ spinners systematically reduces annulus-averaged spin angular velocity, $\langle \Omega(r) \rangle$, and consequently, $\tau_{res(r)}$ (see Fig. 2F and Supplementary Figs. 18–20).

We can now explain why the system self-shears. We observe that while the spinners in the edge layers (labeled $L_1$–$L_3$ in Fig. 2G) are

largely in the registry and move as a cohesive plug, GB scars in layer $L_4$ disrupt layering. This disruption prevents the registry from extending into the bulk, effectively creating a *slip-plane* between the edge and the bulk. The clockwise rotation of spinners in layer $L_3$ exerts a counter-clockwise tangential force on layer $L_4$, and because $\phi_A^{L3} > \phi_A^{L4}$ (Fig. 1D), layer $L_4$ rotates counterclockwise. Spinners in layers further interior fall in the registry again, as the scars have eased the frustration caused by circular confinement. The counterclockwise rotation of layer $L_4$ now couples to this ordered bulk, and the system spontaneously self-shears. On decreasing $\chi$, however, the magnitude of the counter-clockwise tangential force exerted by $L_3$ on $L_4$ also decreases, and self-shearing weakens.

### Odd stresses arising from self-shear drive reentrant melting

With a better understanding of how tuning $\chi$ produces these flows, we now argue that parity-violating inter-spinner collisions generate a radial odd stress ($\sigma_{rr}$), which drives reentrant melting. Let us first consider the case illustrated in Fig. 3A(i), where all particles spin

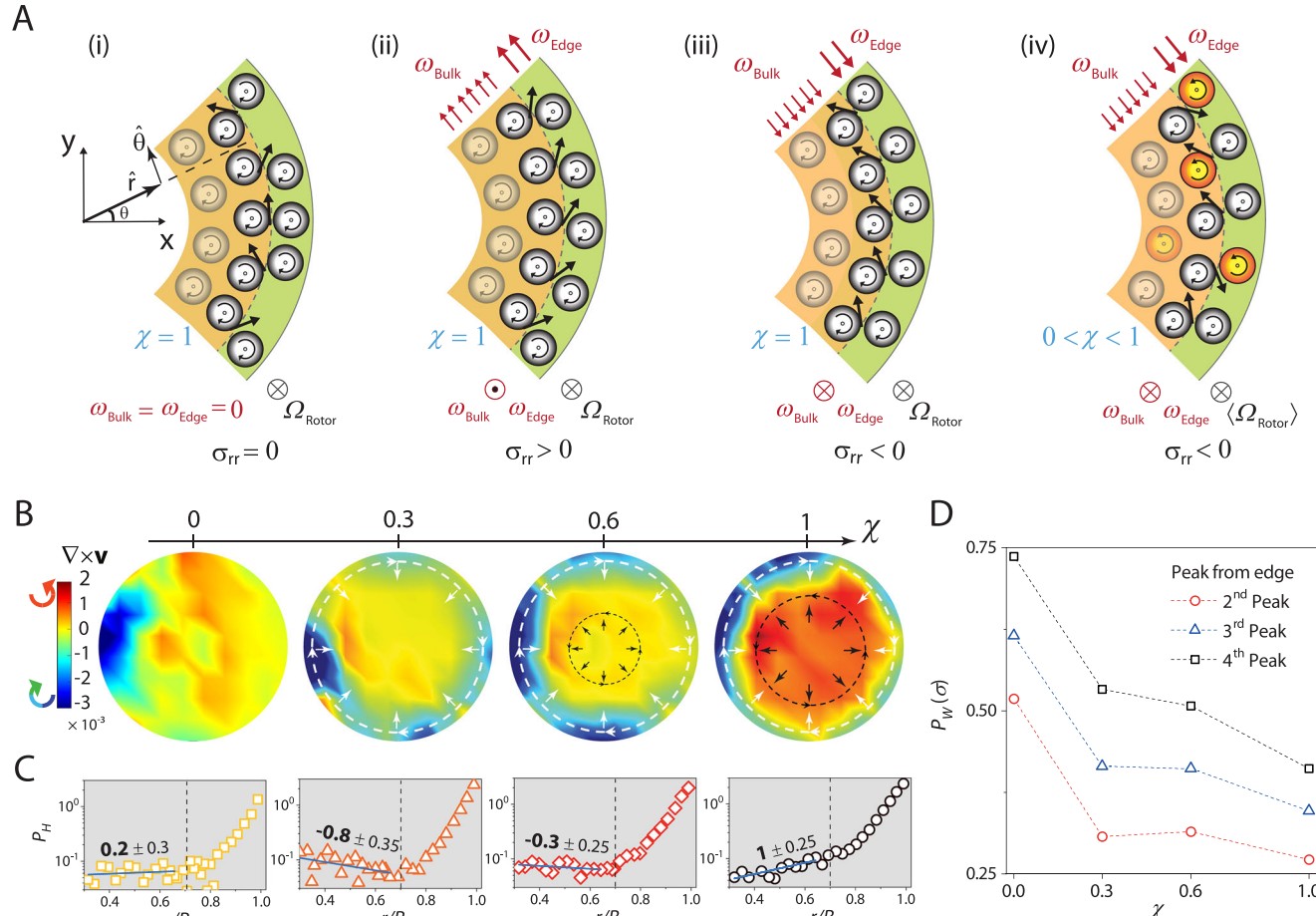

**Fig. 3 | An odd reentrant melting transition. A** Illustration of the microscopic mechanism that gives rise to odd radial stresses in confined spinner materials. The handedness of particle spin, edge, and bulk flows is represented by (⊙) for counterclockwise and (⊗) for a clockwise motion. The green- and orange-shaded regions represent the edge and bulk regions, respectively. Panels (i)–(iii) are for ⊗ spinners. Panel (i): No flow ($\omega_{Edge} = \omega_{Bulk} = 0$). Here, edge spinners collide with bulk spinners from above and below with equal probability. During spinner collisions, the transverse forces exerted by the edge spinners on the bulk ones (black arrows) on average add up to zero along $\hat{r}$ and hence the radial stress $\sigma_{rr} = 0$. Panel (ii): Edge and bulk flow have handedness opposite to particle spin. Edge spinners collide with bulk spinners more frequently from below since $^{\circ}\omega_{Edge} > ^{\circ}\omega_{Bulk}$. The transverse forces now have a component along $\hat{r}$, and, hence, $\sigma_{rr} > 0$. The bulk dilates. Panel (iii): Edge and bulk flow and particle spin have the same handedness, and

$^{\otimes}\omega_{Edge} > ^{\otimes}\omega_{Bulk}$. $\sigma_{rr}$ now points inwards, compressing the bulk. Panel (iv): For $0 < \chi < 1$, although $^{\otimes}\omega_{Edge} > ^{\otimes}\omega_{Bulk}$, transverse forces in many inter-spinner collisions have a component along $\hat{r}$, unlike in Panel (iii), where it is predominantly along $-\hat{r}$. As a result, the stress $|\sigma_{rr}|$ is always smaller than the $\chi = 1$ case. For equal numbers of ⊗ and ⊙ spinners ($\chi = 0$), $\sigma_{rr} = 0$. (**B**), (**C**), and (**D**) are for $\phi = 0.72$. **B** Shows the vorticity obtained from a coarse-grained velocity field for different $\chi$ values. The color bar denotes the magnitude and handedness of the vorticity. The black and the white arrows correspond to $\sigma_{rr}$ along $\hat{r}$ and $-\hat{r}$, respectively. **C** Shows the peak height, $P_H(r)$, of $\phi_A(r)$ for different $\chi$ values. The peak height is measured from the baseline shown in Fig. 1D. The dashed vertical lines delineate the bulk from the edge. **D** Shows the full width at half maximum (FWHM) in units of $d_P$ for peaks close to the edge. The peak width is a direct measure of the total $\sigma_{rr}$.

clockwise (⊗, $\chi = 1$) and bulk and edge flows are absent ($\omega_{Edge} = \omega_{Bulk} = 0$). Here, the edge spinners (green region) are equally likely to collide with bulk spinners (orange region) from above or below. The transverse collision forces from the edge spinners on the bulk spinners (black arrows) average out to zero along $\hat{r}$; only the $\hat{\theta}$ component survives. As a result, $\sigma_{rr} = 0$, and the spinner density is unchanged. In Fig. 3A(ii), the particle spin is still clockwise, but both the edge and bulk flows have a handedness *opposite* to the spin. Since $\omega_{Edge}^{\circ} > \omega_{Bulk}^{\circ}$, edge spinners are more likely to collide with the bulk spinners from below. The transverse forces acquire a net radially outward component. $\sigma_{rr} > 0$, and consequently, the bulk experiences dilation. Conversely, if the handedness of the azimuthal flows is reversed, i.e., it has the *same handedness* as the particle spin with $\omega_{Edge}^{\otimes} > \omega_{Bulk}^{\otimes}$; collisions from above become more frequent (Fig. 3A(iii)). Here, $\sigma_{rr} < 0$, and the bulk is under compression. Importantly, $\sigma_{rr}$ breaks parity and is a consequence of odd material moduli. This behavior should be contrasted with passive granular packings, where dilation-induced normal forces are insensitive to the direction

of shear. Finally, in mixtures of ⊗ and ⊙ spinners, but with the former in excess ($0 < \chi < 1$, Fig. 3A(iv)), ⊗ azimuthal flows still give rise to a compressive radial stress (−ve $\sigma_{rr}$). However, collisions with ⊙ spinners offset this effect and reduce its magnitude, resulting in milder changes in density.

The coarse-grained local vorticity, presented in Fig. 3B for $\phi = 0.72$ at all values of $\chi$, provides a precise representation of the flow field in our system. We calculated the vorticity solely for the longest-lived flows, since short-lived vortices primarily reflect transient fluctuations and do not appreciably affect the steady-state radial density profile $\phi_A(r)$. To directly connect these flow patterns with density changes, we quantified the heights of the radial density peaks, $P_H$, in $\phi_A(r)$ (Fig. 1D) and tracked how $P_H(r)$ varies with $r/R$ (Fig. 3C). A negative slope of $P_H(r)$ in the bulk ($r/R < 0.7$) indicates that this region is compressed, while a positive slope indicates dilation.

At $\chi = 0$, we observed small vortices of both handedness (⊗ and ⊙), but they are too weak to alter the bulk spinner density; as a result, $P_H(r)$ remains essentially flat in the interior ($r/R \leq 0.7$). Further, wall-

induced layering depletes spinners from the bulk, causing its density ($\phi_{\text{Bulk}} \approx 0.706$) to fall below the equilibrium liquid threshold ($\phi_{\text{Eq}}^{\text{Liq}} = 0.716$). The absence of radial stresses for $\chi = 0$ (Fig. 3A(i)) means that the peak widths of $P_H$ near the edge, $P_W$, are broader than at other chiral activities (Fig. 3D). For $\chi = 0.3$, there is now an excess of $\otimes$ spinners. Here, the bulk flow is absent, and the edge flow is clockwise ($\otimes$) (Fig. 2A) and gives rise to a $-\sigma_{rr}\hat{r}$. This stress compacts the bulk, increasing its density beyond the crystallization threshold ($\phi_{\text{Bulk}} \approx 0.725 > \phi_{\text{Eq}}^{\text{Xtal}}$) and drives the bulk into a crystalline state; $|\psi_6|$ is therefore large (Fig. 1C). In addition, due to finite radial stresses, the edge peaks are narrower than those at $\chi = 0$ (Fig. 3D). At larger chiral activity, $\chi = 0.6$ and 1, the situation is more complex since the system self-shears. The inward compressive stress generated by the $\otimes$ edge flow competes with an outward dilative stress from the $\odot$ bulk flow; $P_W$ predictably narrows (Fig. 3D). The contribution from the edge barely wins over and compresses the bulk at $\chi = 0.6$, but this suffices to crystallize the bulk. At $\chi = 1$, the outward radial stress from the bulk overwhelms the inward stress from the edge; the peak widths are at their smallest. The bulk dilates $\phi_{\text{Bulk}} \approx \phi_{\text{Eq}}^{\text{Liq}}$, and the system melts again. Taken together, these results reveal a direct and robust link between odd radial stress and bulk density changes. Notably, this link was present at all $\phi$ and $\chi$ values studied here (Supplementary Figs. 21–23).

## Discussion

In summary, we find that grain boundary scars dramatically alter the behavior of confined active spinner crystals. These scars act as weak links and enable the topologically protected edge flows to decouple from the bulk, triggering spontaneous self-shearing with increasing chiral activity. Parity-violating odd stresses couple the shear flows to density changes, driving the observed reentrant melting transition. That this reentrance is strongest near the equilibrium melting phase boundary (Fig. 1D) is likely no coincidence, as systems in this regime are highly susceptible to phase changes, which can be induced by even small odd stresses. The remarkable robustness of the scar arrangement we observe in both experiments and simulations suggests that the boundary geometry is a powerful tool to control defect placement, offering a new route to engineer flows and induce novel functionalities in odd materials. Our findings pave the way for an exciting future where the intricate interplay of odd elasticity and geometrical frustration unveils physics that is increasingly odd.

## Methods

### Particle fabrication

The spinners were 3D-printed using the PROJET3600 MultiJet 3D printer. The printer has a high resolution of 16 μm, allowing us to print 3D models with small features. These spinners were designed in 3D Builder software. Each spinner has 16 legs attached to the oblate spheroidal dome. We print around 1000 spinners in a batch, which takes about 1–2 h. These spinners are embedded in a wax mold as a part of the print process, and the post-processing requires sonicating them in oil at 60 °C to remove wax. After sonication, the spinners are washed with a soap solution to remove excess oil from the surface.

### Imaging the particles

A high-speed camera was used to image the spinners from the top. The image acquisition rate was between 40 Hz and 30 Hz. These values were chosen appropriately to track the orientation of each spinner. The spatial resolution of 2464 × 2056 pixels was set during data acquisition. The collected data was processed using custom-written code in MATLAB.

### Simulation details

To mimic the experimental system[45,57–63], we simulate a minimal model of a dense collection of spinners in 2D. This granular assembly consists of monodisperse spinners confined in a circular arena. The equation of motion for each spinner, for the position $\mathbf{r}_i$ and angular coordinate $\boldsymbol{\theta}_i$ follows the underdamped equations,

$$m_i\ddot{\mathbf{r}}_i = -\zeta^{\text{T}}\dot{\mathbf{r}}_i + \mathbf{F}_i + \boldsymbol{\eta}_i^{\text{T}} \tag{1}$$

$$I_i\ddot{\theta}_i = \tau_i^{\text{a}} + \tau_i^{\text{int}} + \eta_i^{\text{R}}, \tag{2}$$

where $m_i$ is the mass of the $i$-th spinner, $\zeta^{\text{T}}$ is the translational damping coefficient, $\mathbf{F}_i$ is the net interaction force acting on the $i$-th spinner. This force, $\mathbf{F}_i = \sum_j(\mathbf{F}_{n_{ij}} + \mathbf{F}_{t_{ij}})$ (summed over all neighbors of $i$, indexed by $j$) has two different contributions; one is from a repulsive harmonic normal force $\mathbf{F}_{n_{ij}}$ and the other is from a tangential frictional force $\mathbf{F}_{t_{ij}}$. Also, we compute the interaction torque as

$$\tau_i^{\text{int}} = \sum_{j \neq i} \Gamma_{ij} = -\sum_{j \neq i} \left(\frac{R_i}{R_i + R_j} \mathbf{r}_{ij} \times \mathbf{F}_{t_{ij}}\right) \cdot \hat{\mathbf{z}}. \tag{3}$$

where $\Gamma_{ij}$ is the pairwise frictional torque, $R_i, R_j$ are the radius of spinner $i, j$ respectively, $\mathbf{r}_{ij} = \mathbf{r}_i - \mathbf{r}_j$ and $\hat{\mathbf{z}}$ is the unit vector along $z$ direction. Here, the spinners are distributed in the x–y plane.

To model these forces, we adopt the spring-dashpot model of contact forces (between particle $i$ and $j$) given by

$$\mathbf{F}_{n_{ij}} = -k_n\delta r_{ij}\hat{\mathbf{r}}_{ij} - \zeta_n\mathbf{v}_{n_{ij}} \tag{4}$$

$$\mathbf{F}_{t_{ij}} = -k_t\mathbf{s}_{ij} - \zeta_t\mathbf{v}_{t_{ij}}. \tag{5}$$

Where $k_n$ and $k_t$ are spring constants associated with the normal and transverse harmonic forces. The quantities $\boldsymbol{\delta r}_{ij} = \delta r_{ij}\hat{\mathbf{r}}_{ij}$, $\mathbf{s}_{ij}$ are corresponding displacements for the normal and the transverse direction. Here, $\delta r_{ij} = R_i + R_j - r_{ij}$, the relative distance $r_{ij} = |\mathbf{r}_{ij}| = |\mathbf{r}_i - \mathbf{r}_j|$, and $\hat{\mathbf{r}}_{ij}$ is the unit vector along the relative distance. Normal and tangential damping coefficients $\zeta_n, \zeta_t$ control dissipation during collisions, and $\mathbf{v}_{ij}$ is the relative velocity of the point of contact (where subscripts $n$ and $t$ denote normal and tangential components). We compute these components following,

$$\mathbf{v}_{n_{ij}} = (\mathbf{v}_{ij} \cdot \hat{\mathbf{r}}_{ij})\hat{\mathbf{r}}_{ij} \tag{6}$$

$$\mathbf{v}_{t_{ij}} = \mathbf{v}_{ij} - \mathbf{v}_{n_{ij}} - \left(\frac{R_i}{R_i + R_j}\dot{\theta}_i + \frac{R_j}{R_i + R_j}\dot{\theta}_j\right)(\hat{\mathbf{z}} \times \hat{\mathbf{r}}_{ij}). \tag{7}$$

The transverse displacement $\mathbf{s}_{ij}$ is computed during the period of contact (between spinner $i$ and $j$), i.e., it is initialized at 0 when the contact is just established and integrated numerically following,

$$\dot{\mathbf{s}}_{ij} = \mathbf{v}_{t_{ij}} - \frac{(\mathbf{v}_{t_{ij}} \cdot \mathbf{s}_{ij})\mathbf{r}_{ij}}{r_{ij}^2}. \tag{8}$$

and reset to 0 when the contact is broken. Therefore, the contacts (and correspondingly $\mathbf{s}_{ij}$) are history-dependent and cannot be computed from the instantaneous configuration. Using Eq. (5), we calculate $\mathbf{F}_{t_{ij}}$ and Coulomb's law of contact friction is implemented by limiting $|\mathbf{F}_{t_{ij}}| \leq \mu|\mathbf{F}_{n_{ij}}|$ (after which a contact point starts sliding with constant tangential force).

We also have included a translational noise $\boldsymbol{\eta}_i^{\text{T}}$ whose mean is zero $\langle\boldsymbol{\eta}_i^{\text{T}}(t)\rangle = 0$ and its variance is given by $\langle\eta_{i,\alpha}^{\text{T}}(t)\eta_{j,\beta}^{\text{T}}(t')\rangle = 2D^{\text{T}}\delta_{ij}\delta_{\alpha\beta}\delta(t - t')$ where $D^{\text{T}}$ represents the translational diffusion constant associated with this noise.

For the equation describing the angular degrees of freedom, $I_i$ is the moment of inertia and $\tau_i^a$ is the active torque, modeled as

$\tau_i^a = \alpha(\omega_0 - \omega_i)$ where $1/\alpha$ is the relaxation time scale and $\omega_i = \frac{d\theta_i}{dt}$. Similarly, as before, for the angular degrees of freedom we have a rotational noise $\eta_i^R$ whose mean is zero $\langle \eta_i^R(t) \rangle = 0$ and its variance is given by $\langle \eta_i^R(t)\eta_j^R(t') \rangle = 2D^R \delta_{ij}\delta(t - t')$ where $D^R$ represents the rotational diffusion constant associated with this noise.

We use $N_b = 800$ frozen (whose position, momentum, and angular degrees of freedom do not evolve) particles to create circular confinement. Unless stated otherwise, we use the size of the circular confinement $r_0 = 20.60$, mass $m_i = 7.0$, the radius of each spinner $R_i = R_0 = 0.288$, $\zeta^T = 0.15$, $D^T = 4 \times 10^{-5}$, $D^R = 10^{-3}$, $\omega = 20$, $\alpha = 100$, $\zeta_n = 4.0$, $\zeta_t = 2.0$, $k_n = 100$, $k_t = 100$. Between all the mobile spinners, we used a friction coefficient $\mu_p = 0.60$, and between the mobile spinners and the frozen boundary particles, we used a friction coefficient $\mu_p = 0.65$. For different area fractions, we either use (a) different number of spinners, i.e., $N = 3689$ for $\phi = 0.72$, $N = 4050$ for $\phi = 0.79$, etc., or (b) fixed number of spinners $N = 3689$ with different spinner radius $R_0 = 0.279$, 0.288, 0.293, and 0.301 for $\phi = 0.68$, 0.72, 0.75, 0.79. To explore the system size effects for the passive disks, we performed simulations with $N = 500$, 1000, and 2000 as well. To analyze the radial profile of different observables, we divided the circular arena into $n_r = 40$ annuli and averaged the observables over the spinners inside each annulus.

## Data availability
The source data files for the main figures and supplement generated in this study have been deposited in the Figshare.com database under accession code 10.6084/m9.figshare.30959399.

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

## Acknowledgements

U.T. thanks the Jawaharlal Nehru Centre for Advanced Scientific Research, Bangalore, India, for a research fellowship. P.A. thanks the Jawaharlal Nehru Centre for Advanced Scientific Research, Bangalore, India, for a research fellowship. A.K.S. thanks the Science and Engineering Research Board, Government of India, for the National Science Chair. S.R. acknowledges a J C Bose Fellowship of the ANRF, INDIA. R.M. acknowledges support from the ANRF, India, through PMECRG (project ANRF/ECRG/2024/002036/PMS). R.G. thanks the Department of Science and Technology, India, for financial support through the Swarnajayanti Fellowship Grant (DST/SJF/PSA-03/2017-22). U.T. and R.G. thank Shreyas Gokhale for critical inputs on the manuscript.

## Author contributions

U.T. and R.G. planned and executed experimental research and designed and performed analysis. R.M. performed simulations. P.A. contributed to project design and trained U.T. on experimental methodology. A.K.S. and S.R. contributed to project development. U.T., R.M., and R.G. wrote the paper with inputs from all authors.

## Competing interests

The authors declare no competing interests.
