## [Transparent Peer Review file · Nature Communications]

Reentrant melting of scarred odd crystals by self-shear

Corresponding Author: Mr Uttam Tiwari

Version 0:

Reviewer comments:

Reviewer #1

(Remarks to the Author)

I apologise for the delay in my report. In this article, Tiwari et al construct an experimental system of confined spinners with a mixture of orientations, showing that they exhibit odd self-rheology characterised by a coexistence between crystalline regions and regions with defect scars near the edge. The chiral, self-shearing motion is concentrated in these regions and is sensitive to local and global packing fraction, and to the relative concentration of spinners of different orientations.

I find the research carefully carried out, and the results interesting. However, my main comment is that the current format with an exceptionally short main text makes the arguments rather vague and difficult to follow.

I would support publication if the authors satisfactorily answer my questions below and also if they rewrite their manuscript to include substantial parts of the supplementary in the main.

Specific comments:

Edge flows, self shear direction and reentrant melting: Clearly, there is something complex going on here, and I don't think there is a full explanation in the manuscript.

The authors interpret their results within the framework of continuum odd elasticity, which has in recent years been developed for systems with internal chiral torques. That is of course fine, but it does not capture jamming / unjamming or solid / liquid transition in the system.

The author observe the scars, and link them to lowered chiral active stresses in this region because of fewer contacts that transfer torques due to lower density, as well as directly through simulations. The authors develop an intricate microscopic interpretation of the torques near the grain boundaries. I could not make sense of Figure 3A and the inferred torque and stress directions, and why some systems rotate counterclockwise. The link between torque directions and the odd moduli coupling (equation Fig. S16) is also pretty hand-waivy.

In the manuscript, there is an extremely brief reference to 'simulations', with data shown only in Figure 2C and 2E. In the supplementary, it then emerges that this is a detailed molecular dynamics simulation with Coulomb friction, essentially identical to granular materials models except for the inclusion of the spinning torque.

This then also allows for a better computation of the chiral torques compared to the 'frictional collisions' of [37], with a dip at the scar (Fig 2E). To me a proper introduction of the simulation belongs in the manuscript directly.

It should also allow for calculating the stresses and their profiles directly? In the manuscript, the authors relate compression / dilation to the peak height of packing fraction peaks, which seems rather indirect and which I've never seen before. Is there not an even approximate pressure / density constitutive law that one could infer here?

Seen from the granular angle, what the authors have here is to me a shear band, very similar to the ones seen in (driven) granular materials in a couette shear cell geometry (see e.g. Schall P, Van Hecke M. Shear bands in matter with granularity. Annual Review of Fluid Mechanics. 2010;42(1):67-88.).

The core feature is an instability triggered by an inversion in the (shear) stress / strain rate curve, which then makes the system separate into jammed and flowing regions. I suspect that this concept could be usefully translated to the active case,

possibly with the net internal oddness χ as a tuning parameter of the instability?

There is a well developed literature on models of shear bands that take into account density, friction, and constitutive laws which include nonlocal effects on stress (see e.g. S. Mandal et al, Phys. Rev. X 11, 021017 (2021)). It would relate stresses, density and strain rates / vorticity, like the manuscript tries to do.

Can the authors relate their observations on strain rate inhomogeneities (\sim vorticity), density fluctuations and stresses to this framework, even tentatively?

All in all, after reading this manuscript I feel I still don't understand why this phenomenon happens.

Reviewer #2

(Remarks to the Author)

In this work, the authors investigate the melting behavior of confined active spinners under varying activity. Combining experiments and simulations, they report a reentrant melting phenomenon driven by the interplay between odd response, edge flow and confinement-induced defects, where the crystalline order of the bulk is higher at intermediate activity (compared to vanishing mean activity) but is disrupted at larger activity at a fixed spinner density. The finding is novel, and the work is well-organized. Given the current growing interest in chiral active matter and odd response, the manuscript should be seriously considered for publication in NC. I will be happy to review the manuscript once the authors will have addressed my points in the following. However, as for now, I am still not in the position to recommend publication of the manuscript.

1. In this study, the system is binary mixture of clockwise and counter-clockwise spinners, and the chiral activity is changed by tuning the molar fraction of clockwise spinners. A more direct way to change activity is to use monodisperse spinner and then to tune the intensity of vertical vibration. Why do the authors not employ such monodisperse system? In addition, in experiments, what is the initial configuration; and in the steady state, are the clockwise and counter-clockwise spinners uniformly mixed?

2. On page 5, the authors state that "the magnitude of the edge flow drops precipitously at $r/R \approx 0.9$. This behavior is distinct from observations made hitherto in spinner liquids and solids." Its structural origin looks to be the existence of the grain boundary scar near this location. Further, the authors describe that "Unlike conventional grain boundaries, which terminate at the system edge, these scars terminate within the system." An interesting question is why the grain boundary scars emerge at this specific location (around the 4th layer)? Did the authors perform equilibrium simulation of passive disks (with frictional or smooth boundary) in a circular confinement to examine the position distribution of the GBS? Moreover, does the "self-shear" and "scarred defects" depend on the curvature of the confinement (finite-size effect)?

3. In Supplementary Figure 18, the angular velocity profile at $\phi = 0.75$ is non-monotonic, differing significantly from those at other packing fractions. Additionally, the profile at $\phi = 0.75$ appears to display a weak self-shear instead of vanishing self-shear. The authors should provide further discussion on these observations.

4. The data for annular angular velocities (Fig. 2a) and spin velocities (Fig. 2f) exhibit considerable fluctuations. To enhance the data quality, more extensive statistical averaging should be conducted.

5. The strengths of reentrance Δ plotted in Fig.2c and Fig.2d appear to have different meanings. And, they are not equivalent with the definition in the second paragraph of page 4. Please carefully check this.

6. A typo: Page 7, second paragraph: "spinners in layers further interior fall in..." should read "Spinners in layers further interior fall in..."

Version 1:

Reviewer comments:

Reviewer #2

(Remarks to the Author)

I have reviewed the revised manuscript. The authors have basically addressed my questions and comments, while parts of which are not quite satisfactorily addressed. Given that this work is sufficiently interesting, I would like to recommend its publication in Nature communications.

Reviewer #3

(Remarks to the Author)

In this manuscript, the authors study a 2D mixture of granular spinners with two different chiralities. By varying the ratio of the two chiralities and their overall density, the authors observe the emergence of a large-scale single crystal at intermediate values of the net chiral activity, in contrast to liquid-like regions which appear at large and small values. Combining

experimental observations with simulations, the authors demonstrate that confinement-induced defects are responsible for decoupling the bulk and edge rotations, providing detailed insights into the mechanisms behind the observed structural and dynamical behaviours.

I believe the authors have addressed the previous reviewers' comments in detail, and the work presents a novel and timely contribution. One of the strength of the work is the combination of experiments and simulations that demonstrate the range of the phenomena in the manuscript. However, I suggest the authors further edit the manuscript for clarity.

Comment 1:

The data in Fig.~1D exhibits only modest variation, with significant error bars. The authors claim that " Δ is non-monotonic with ϕ , with a maximum at $\phi = 0.72$," but I am not convinced the figure supports this statement. I suggest a rephrasing to " Δ appears to be non-monotonic with ϕ , with an apparent maximum near $\phi \approx 0.72$." I also recommend increasing the thickness and contrast of the error bars for clarity.

Comment 2:

The sentence "We calculated vorticity solely for the longest-lived flows since only these can significantly impact the radial density profile" is not clear to me and should be rephrased. Why do short-lived vortices not impact $\phi_A(r)$?

Comment 3:

The Methods introduce translational and rotational diffusion coefficients (D^T , D^R), but their physical origin is not clear. Could the authors clarify the experimental source of these terms and how their values were chosen?

Comment 4:

Supplementary Fig.~S1 shows the distribution of same-type nearest neighbours at $\chi=0, \phi=0.72$. The authors should clarify whether these distributions were computed over the entire system or only the bulk region.

Reviewer #4

(Remarks to the Author)

Response to reviewer's comments

Reviewer #1 (Remarks to the Author):

Comment: I apologise for the delay in my report. In this article, Tiwari et al construct an experimental system of confined spinners with a mixture of orientations, showing that they exhibit odd self-rheology characterised by a coexistence between crystalline regions and regions with defect scars near the edge. The chiral, self-shearing motion is concentrated in these regions and is sensitive to local and global packing fraction, and to the relative concentration of spinners of different orientations.

I find the research carefully carried out, and the results interesting. However, my main comment is that the current format with an exceptionally short main text makes the arguments rather vague and difficult to follow. I would support publication if the authors satisfactorily answer my questions below and also if they rewrite their manuscript to include substantial parts of the supplementary in the main.

Response: We thank the referee for recognizing that our study is carefully carried out and for finding the results interesting. We are also grateful for the constructive feedback. We apologize for the delay in submitting this revision. Our instrument was committed to another set of measurements that had to be completed before we could perform the experiments needed to address the comments raised by Reviewer 2.

Below, we provide a detailed point-by-point response. All changes made in the main text and Supplementary Information are highlighted in magenta.

Comment 1 : Edge flows, self shear direction and reentrant melting: Clearly, there is something complex going on here, and I don't think there is a full explanation in the manuscript.

Response 1: As the referee points out, there is an intricate interplay between self-shear-generated odd stresses, which depend on the chiral activity, and reentrant melting. In addition, grain boundary scars are responsible for the edge-bulk decoupling. We agree with the referee that the manuscript will benefit from a more detailed explanation. To this end, we have rewritten substantial portions of the results section in our revision, as well as improved the overall readability of the manuscript. Please see all the highlighted sections. If the referee feels additional information is required, we will include it.

Comment 2: The authors interpret their results within the framework of continuum odd elasticity, which has in recent years been developed for systems with internal chiral torques. That is of course fine, but it does not capture jam-

ming/unjamming or solid/liquid transition in the system.

Response 2: The referee is correct. Odd elasticity is a relatively recent framework (Ref. [54]) and is yet to be applied to the specific system we present. Importantly, we were able to observe chiral-activity-driven reentrant melting (Fig. 1B) only because our system operated near an equilibrium liquid–solid boundary ($\phi = 0.72$, Ref. [51]). Thus, small changes in the control parameter—the net chiral activity χ —suffice to drive the system across this boundary. Our explanation of the reentrant melting phenomenon relies on a few key concepts that were put forth in Refs. [38,54]: in materials with internal torque density, parity violation gives rise to odd elastic moduli, such that azimuthal flows can generate radial stresses. The sign of these stresses determines whether the material dilates or compresses. Our experiments demonstrate a causal link between vorticity and density changes—a unique hallmark of odd materials.

Comment 3: The author observe the scars, and link them to lowered chiral active stresses in this region because of fewer contacts that transfer torques due to lower density, as well as directly through simulations. The authors develop an intricate microscopic interpretation of the torques near the grain boundaries. I could not make sense of Figure 3A and the inferred torque and stress directions, and why some systems rotate counterclockwise. The link between torque directions and the odd moduli coupling (equation Fig. S16) is also pretty hand-waivy.

Response 3: We apologize for the earlier lack of clarity. To illustrate the dependence of stress directions on the handedness of azimuthal flows, we can first consider a dense passive granular assembly under boundary shear: such a system dilates and produces outward normal stresses, regardless of shear direction. In contrast, for spinning particles, azimuthal flows are *self-generated*, and inter-spinner collisions break parity. Thus, the radial stress direction depends on whether the flow handedness aligns or opposes the intrinsic particle spin. In Fig. 3A, we now present these cases more clearly, with revised explanations and captions (see lines 209–228 and lines 231–254).

Supplementary Fig. S15 schematically illustrates how odd elastic moduli couple strain and strain rates to stresses (adapted from Ref. [6] of the Supplement, or [44] of the main paper). For example, in a system of clockwise spinners, a finite K_{\perp}^1 means clockwise edge flows result in compressive stresses, while counterclockwise ones result in dilational stresses. It is these odd coefficients that are also at play in our experiments that result in the observed reentrant melting behavior.

Comment 4: In the manuscript, there is an extremely brief reference to 'simulations', with data shown only in Figure 2C and 2E. In the supplementary, it then emerges that this is a detailed molecular dynamics simulation with Coulomb friction, essentially identical to granular materials models except for the inclusion of the spinning torque. This then also allows for a better computation of

the chiral torques compared to the 'frictional collisions' of [37], with a dip at the scar (Fig 2E). To me a proper introduction of the simulation belongs in the manuscript directly.

Response 4: We appreciate this suggestion. In our revised manuscript, we introduce the simulations immediately after introducing the experimental system (line nos: 87-93). The complete simulation details are now included in the Materials and Methods section, which is now part of the main manuscript. If the referee would like us to include further details in the main text, we will do so.

Comment 5: It should also allow for calculating the stresses and their profiles directly? In the manuscript, the authors relate compression / dilation to the peak height of packing fraction peaks, which seems rather indirect and which I've never seen before. Is there not an even approximate pressure / density constitutive law that one could infer here?

Response 5: We thank the referee for this comment. In our experiments, we have access only to particle positions and not the forces. Hence, we use the peaks of the radial density profile $\phi_A(r)$ as an indirect measure of the pressure. In regions where the pressure is large, the rattle room available for the particle is small, and hence the peaks grow in height and shrink in width. We would also like to note here that for active matter, in general, pressure is not a state function. Even for the case of achiral active matter, the pressure, for instance, depends on the wall curvature and roughness. This arises from the fact that active particles are persistent, and hence the wall features that promote/suppress particle alignment modify the pressure, and it is not just dependent on the bulk particle density (see Solon et al., Nature Phys. 11, 673 (2015)). For chiral active matter again, particles that rotate with a well-defined handedness hug walls due to intrinsic torques [Barois et al., Phys. Rev. Lett. **125**, 238003 (2020) and Kant et al., <https://arxiv.org/abs/2509.00729>], and we expect pressure to depend on the wall properties. When we consider the case of spinners, the ubiquitous edge flows near a boundary will generate odd radial stresses, which will contribute to the pressure at the wall. The direction of these flows, and hence the pressure, depends on inter-spinner and spinner wall friction. This makes inferring a pressure-density constitutive law difficult.

Nonetheless, since in our simulations we have access to both the positions and forces, we have attempted to calculate whether the pressure and density are correlated. We calculate the local virial pressure, $p_i = -\frac{1}{A} \sum_{\substack{j=1 \\ j \neq i}}^N (x_{ij} F_{ij}^x + y_{ij} F_{ij}^y)$ to relate pressure to the density changes in our system. Here, A is the Voronoi cell area obtained from the Voronoi tessellation of the spinners. The quantities $x_{ij} = x_i - x_j$ and $y_{ij} = y_i - y_j$ represent the Cartesian distance between the i^{th} and j^{th} spinner along x and y directions. The terms F_{ij}^x and F_{ij}^y represent, respectively, the x and y components of forces acting on the i th spinner due to the j th spinner. Here, N is the number of nearest neighbors. In

Fig. 1A, we show the annular virial pressure, P , and annular area fraction ϕ_A at $\chi = 1$ and $\phi = 0.72$. Consistent with intuition and our observations (Line nos: 229-235), the pressure and density are clearly correlated. In addition, the drop in density in the annulus harboring the grain boundary scars is accompanied by a drop in the pressure as well. In Fig. 1B, we also show the dependence of the virial pressure, P , on the spinners versus Voronoi cell area of the spinners, A , to demonstrate the relationship between local pressure and local area in our system. If the referee wants us to include this figure in the supplement, we will do so.

Fig. 1: (A) Pressure (black squares) and density (red squares) versus r/R in simulations for $\chi = 1$ at $\phi = 0.72$. (B) Pressure versus Voronoi cell area for $\chi = 1$ at $\phi = 0.72$.

Comment 6: *Seen from the granular angle, what the authors have here is to me a shear band, very similar to the ones seen in (driven) granular materials in a couette shear cell geometry (see e.g. Schall P, Van Hecke M. Shear bands in matter with granularity. Annual Review of Fluid Mechanics. 2010;42(1):67-88.).*

Response 6: We thank the referee for this interesting perspective. We acknowledge that certain findings of ours invite comparison with shear banding in driven granular materials. For instance, in boundary-driven granular matter, shear bands nucleate in regions where a local yield threshold is crossed. In our experiments, the slip between the edge and bulk flows indeed occurs in these poorly packed regions where the system is more susceptible to yield. Having said that, we note that in banded flows in driven passive granular materials, the width of the band is set by non-local terms in the constitutive law. In our experiments, in contrast, it is purely determined by the location of the scars, which is controlled by boundary geometry.

We refer to the phenomenon as self-shear rather than a shear band because it arises *spontaneously* without boundary forcing. The boundary, here, is static. Furthermore, in boundary-driven passive granular systems, the material response, which includes dilation-induced stresses, does not depend on the shear

direction - *all stresses are parity-even* (see response to comment 3). In materials where the constitutive building blocks spin with a well-defined handedness, since the microscopic interactions are parity-violating, this is not the case. These distinctions give rise to odd material moduli, and make our findings distinct from boundary-driven granular shear bands.

Comment 7: The core feature is an instability triggered by an inversion in the (shear) stress/strain rate curve, which then makes the system separate into jammed and flowing regions. I suspect that this concept could be usefully translated to the active case, possibly with the net internal oddness as a tuning parameter of the instability?

Response 7: Unlike in generic shear banding systems where one observes jammed and flowing regions, in our experiments the edge region rotates as a cohesive plug and the bulk region in the immediate vicinity of the scar ($0.8 < r/R < 0.9$) is also quite well ordered (Fig. 1B) and rotates as a plug, but with opposite handedness, for $\chi = 1$ (Fig. 2A). The scar region acts as a slip plane between two solid-like regions. It is the outward radial stress due to counter-rotating bulk that causes the bulk to melt for $r/R < 0.7$. In fact, for the smaller chiral activity values ($\chi = 0.3$ and 0.6), the solid-like bulk and edge regions are more apparent.

The referee is indeed correct that net internal oddness, or chiral activity, tunes the edge and bulk flows. But whether this can be mapped to the inversion seen in stress versus strain curves in systems undergoing shear banding is not clear, and is definitely worth investigating in future studies. The standard shear banding instability often incorporates a flow-structure feedback mechanism. One key difference that does arise in our system is that on systematically increasing χ , we observe a reentrant change in the structure. To the best of our knowledge, we are not aware of such a feature arising in systems undergoing shear banding.

Comment 8: There is a well-developed literature on models of shear bands that take into account density, friction, and constitutive laws which include nonlocal effects on stress (see e.g. S. Mandal et al, Phys. Rev. X 11, 021017 (2021)). It would relate stresses, density and strain rates / vorticity, like the manuscript tries to do. Can the authors relate their observations on strain rate inhomogeneities (vorticity), density fluctuations and stresses to this framework, even tentatively?

Response 8: We thank the referee for pointing us to the above paper and the non-local granular rheology literature. The mechanism that we invoke to explain our observations hinges on odd elasticity theory [Ref. 54]. This framework provides a local constitutive law between stress and strain (Supp. Fig. 15). When this is combined with the force-balance condition $\nabla \cdot \sigma + \text{substrate drag} = 0$ it becomes a boundary-value problem, which produces non-local effects: vorticity in one region affects the density and stress elsewhere. This suffices to qualita-

tively explain all our results. We emphasize that our primary goal here was to demonstrate that odd stresses alone can drive phase changes in materials, and our experiments and simulations show this can be achieved.

In non-local granular rheology models, the non-locality is built into the constitutive law itself by introducing a flow screening length. This presents many advantages; for instance, it allows for tuning the width of shear bands or capturing far-field creep. Odd mechanics is itself a very recent development (Ref. [54]). Building non-locality into the constitutive laws for systems with parity-odd stresses is an exciting theoretical challenge, but it is beyond the scope of the present work.

Comment 9: All in all, after reading this manuscript I feel I still don't understand why this phenomenon happens.

Response 9: We thank the referee for this constructive remark. To improve clarity, we have substantially revised sections of the manuscript. We believe these revisions have enhanced the overall readability. We hope the referee is satisfied with the changes made.

Reviewer #2 (Remarks to the Author):

Comment : In this work, the authors investigate the melting behavior of confined active spinners under varying activity. Combining experiments and simulations, they report a reentrant melting phenomenon driven by the interplay between odd response, edge flow and confinement-induced defects, where the crystalline order of the bulk is higher at intermediate activity (compared to vanishing mean activity) but is disrupted at larger activity at a fixed spinner density. The finding is novel, and the work is well-organized. Given the current growing interest in chiral active matter and odd response, the manuscript should be seriously considered for publication in NC. I will be happy to review the manuscript once the authors will have addressed my points in the following. However, as for now, I am still not in the position to recommend publication of the manuscript.

Response: We thank the referee for finding our study novel and the work well-organized. We also thank the referee for their constructive comments. We apologize for the delay in submitting our revision. Our instrument was committed to another set of measurements that had to be completed before we could perform the experiments needed for addressing this referee's comments.

Below, we provide a point-by-point response. All the changes made to the text in the main manuscript and supplement are colored magenta.

Comment 1: In this study, the system is binary mixture of clockwise and counter-clockwise spinners, and the chiral activity is changed by tuning the mo-

lar fraction of clockwise spinners. A more direct way to change activity is to use monodisperse spinner and then to tune the intensity of vertical vibration. Why do the authors not employ such monodisperse system? In addition, in experiments, what is the initial configuration; and in the steady state, are the clockwise and counter-clockwise spinners uniformly mixed?

Response 1: We thank the referee for this question. While tuning the vertical vibration amplitude is a potential approach to control activity in homochiral systems, it has limitations. Specifically, reaching $\chi = 0$ requires turning off the vibration, which halts dynamics entirely due to the absence of thermal motion, preventing comparison with passive systems. Moreover, it was not evident a priori whether a racemic mixture ($\chi = 0$) would behave like vibrated passive disks. We also observed that our particles spin consistently only within a limited range of vibration amplitudes and frequencies.

Our experimental system is assembled by hand and we tried to ensure uniform mixing of clockwise and counterclockwise spinners prior to each experiment. To test whether the mixture exhibited a tendency to phase-separate, we analyzed the local distribution of spinner types for the $\chi = 0$, $\phi = 0.72$ system. Specifically, we measured the number of clockwise-spinning nearest neighbors surrounding a clockwise spinner, and likewise for counterclockwise spinners, at both the beginning and end of the experiment (Supplementary Fig. 1). The resulting distribution is sharply peaked at three and remains unchanged over time. Given that the local coordination number in our experiments is six, this outcome indicates that each particle is, on average, surrounded by equal numbers of clockwise and counterclockwise spinners.

Comment 2: On page 5, the authors state that “the magnitude of the edge flow drops precipitously at $r/R \approx 0.9$. This behavior is distinct from observations made hitherto in spinner liquids and solids.” Its structural origin looks to be the existence of the grain boundary scar near this location. Further, the authors describe that “Unlike conventional grain boundaries, which terminate at the system edge, these scars terminate within the system.” An interesting question is why the grain boundary scars emerge at this specific location (around the 4th layer)? Did the authors perform equilibrium simulation of passive disks (with frictional or smooth boundary) in a circular confinement to examine the position distribution of the GBS? Moreover, does the “self-shear” and “scarred defects” depend on the curvature of the confinement (finite-size effect)?

Response 2: We thank the referee for this comment. The spatial configuration of GB scars in disk packings in 2D remains an open problem and is closely related to Thomson’s packing problem on a sphere. We also do not know as yet if friction between disks alters the scar configuration. Having said that, a plausible explanation for our observations is the following. By expelling the GB scars near to the boundary, frustration in the bulk is relieved and the system can adopt defect-free crystalline configurations, where the rattle room available per particle

is large. This is akin to the crystallization of hard spheres, which is driven by entropy, especially that arising from vibrational contributions. However, if the scars are pushed all the way to the periphery, frustration from curvature can still penetrate into the bulk. The trade-off between these effects, in all likelihood, results in the observed scar configuration.

Following the referee’s suggestion, we performed equilibrium simulations of passive disks under circular confinement at $\phi = 0.72$ for different extents of confinement (system sizes). Voronoi tessellations reveal azimuthally aligned scars near the boundary for $N = 2000$ and $N = 1000$, but less so for $N = 500$ (Supplementary Fig. 11). We also see defects in the bulk for $N = 2000$ across different runs. Importantly, like in the case of particle packings on a sphere (Ref. 9 and 14 of the main manuscript), the scar length decreases with increasing curvature.

To probe the impact of curvature of confinement on self-shear and scar configuration, we have performed additional experiments for different system sizes: $N = 2000$, 1000, and 500 particles at $\phi = 0.72$ and $\chi = 1$. In all these experiments we observed azimuthally-aligned GB scars near the boundary at $r/R \approx 0.9$ (Supplementary Fig. 9). Also, we observed self-shearing for $N = 2000$ particles, but not for the other system sizes. For $N = 1000$ and $N = 500$, we nonetheless see a sudden drop in $\omega(r)$ around $r/R = 0.9$; there is still a decoupling between the edge and the bulk mediated by GB scars. But for these smaller system sizes, the bulk rotates in the same direction as the edge but with a smaller angular velocity.

Comment 3: In Supplementary Figure 18, the angular velocity profile at $\phi = 0.75$ is non-monotonic, differing significantly from those at other packing fractions. Additionally, the profile at $\phi = 0.75$ appears to display a weak self-shear instead of vanishing self-shear. The authors should provide further discussion on these observations.

Response 3: We note that our simulations are a minimal model and do not capture all the subtleties of the experimental observations. The primary objective of the simulation was to determine if the experimentally observed scar configuration could be reproduced in our model and then probe its impact on the torque transfer between layers. The dip in torque transfer in the annulus harboring GB scars supports our arguments (Fig. 2E and Supplementary Fig. 17). At the same time, features such as the lift-off of $\omega(r)$ for all values of ϕ for $r/R < 0.6$ seen in simulations arise from features that we do not fully understand yet. Possible reasons could be that in our experiments, the particles are dome-shaped particles, where some overlap is possible, not disks like in the simulations. The particles are also confined between a top glass plate and a bottom aluminum plate with different friction coefficients. In addition, our system is quasi-2D and not perfectly 2D like in the simulations. The simulations also assume that the spin speeds of all particles are identical. This can never be the case in experiments.

The experimental findings reported in the first submission are strengthened by new experiments we report for different system sizes (Supplementary Fig. 9). In fact, for $N = 2000$ particles, we have reproduced the self-shearing phenomenon at $\chi = 1$. These new experiments are much longer runs (30 min duration) and we still do not observe a lift-off in $\omega(r)$ for $r/R < 0.6$ as seen in the simulations.

Comment 4: The data for annular angular velocities (Fig. 2a) and spin velocities (Fig. 2f) exhibit considerable fluctuations. To enhance the data quality, more extensive statistical averaging should be conducted.

Response 4: Following the referee’s comment, we revisited these data, particularly looking for the origin behind these large fluctuations in the annular angular and spin velocities. Our new analysis, presented in Supplementary Fig. S20A and B, revealed that the fluctuations in the annular spin velocity have a one-to-one correspondence with the variations in net chirality within each annulus. Hence, these fluctuations have an underlying structural origin - the distribution of clockwise and counterclockwise spinners within each annulus - and cannot be averaged away. This also explains why $\Omega(r)$ for $\chi = 1$ (Fig. 2f) is relatively smooth in comparison.

For the annular angular velocity, we once again found a strong correspondence with the annular density fluctuations (Supplementary Fig. S20C, for $\chi = 1$ and $\phi = 0.72$). Here, however, this correspondence is more pronounced for $r/R > 0.6$. For $r/R \leq 0.6$, where there are many defects present, $\omega(r)$ shows large fluctuations (circles in Fig. 2A).

Comment 5: The strengths of reentrance Δ plotted in Fig.2c and Fig.2d appear to have different meanings. And, they are not equivalent with the definition in the second paragraph of page 4. Please carefully check this.

Response 5: We apologize for this oversight. The erroneous depiction of Δ in Fig.1C has been removed. The definition for strength of reentrance, $\Delta = \frac{\langle |\psi_6| \rangle_{\chi=0.3}}{\langle |\psi_6| \rangle_{\chi=1}}$, has been incorporated in the revision, which aligns with Fig. 1D.

Comment 6: A typo: Page 7, second paragraph: “spinners in layers further interior fall in...” should read “Spinners in layers further interior fall in...”.

Response 6: We have now corrected this in our revision.

We hope that the referee finds our responses satisfactory and that the manuscript is suitable for publication in Nature Communications.

Response to reviewer’s comments

Reviewer #2 (Remarks to the Author):

Comment: I have reviewed the revised manuscript. The authors have basically addressed my questions and comments, while parts of which are not quite satisfactorily addressed. Given that this work is sufficiently interesting, I would like to recommend its publication in Nature communications.

Response: We thank the referee for finding our manuscript suitable for publication in Nature Communications.

Reviewer #3 (Remarks to the Author):

Comment : In this manuscript, the authors study a 2D mixture of granular spinners with two different chiralities. By varying the ratio of the two chiralities and their overall density, the authors observe the emergence of a large-scale single crystal at intermediate values of the net chiral activity, in contrast to liquid-like regions which appear at large and small values. Combining experimental observations with simulations, the authors demonstrate that confinement-induced defects are responsible for decoupling the bulk and edge rotations, providing detailed insights into the mechanisms behind the observed structural and dynamical behaviours.

I believe the authors have addressed the previous reviewers’ comments in detail, and the work presents a novel and timely contribution. One of the strength of the work is the combination of experiments and simulations that demonstrate the range of the phenomena in the manuscript. However, I suggest the authors further edit the manuscript for clarity.

Response: We thank the referee for finding that we have addressed the earlier reviewers’ comments in a thorough manner and that the work offers a novel and timely contribution.

Below, we provide a detailed point-by-point response.

Comment 1 : The data in Fig. 1D exhibits only modest variation, with significant error bars. The authors claim that “ Δ is non-monotonic with ϕ , with a maximum at $\phi = 0.72$,” but I am not convinced the figure supports this statement. I suggest a rephrasing to “ Δ appears to be non-monotonic with ϕ , with an apparent maximum near $\phi \approx 0.72$.” I also recommend increasing the thickness and contrast of the error bars for clarity.

Response 1: We thank the referee for this comment. In our revised manuscript,

we have rephrased the statement to: “ Δ appears to be non-monotonic with ϕ , with an apparent maximum near $\phi \approx 0.72$.” **Please see lines 110–111 of the revised main manuscript.**

Additionally, we have increased the thickness and contrast of the error bars in Fig. 1D to enhance visual clarity.

Comment 2 : The sentence “We calculated vorticity solely for the longest-lived flows since only these can significantly impact the radial density profile” is not clear to me and should be rephrased. Why do short-lived vortices not impact $\phi_A(r)$?

Response 2: We thank the referee for raising this point. Using the coarse-grained vorticity, we aimed to depict the flow field that is primarily responsible for affecting the *steady-state* radial density profiles shown in Fig.1E. Short-lived vortices may correspond to localized flows that exist only in limited regions of the system. These short-lived vortices primarily reflect short-term fluctuations in the emergent bulk and edge flows; therefore, they do not accurately represent the global emergent flows and, consequently, the steady-state density profile (see Ref. [39] of Main Manuscript). However, long-lived vortices correspond to large, system-spanning vortices that accurately capture the flow field computed over the experimental duration. We have now rephrased the sentence to: “We calculated the vorticity solely for the longest-lived flows, since short-lived vortices primarily reflect transient fluctuations and do not appreciably affect the steady-state radial density profile $\phi_A(r)$ ”.

Comment 3 : The Methods introduce translational and rotational diffusion coefficients (D^T , D^R), but their physical origin is not clear. Could the authors clarify the experimental source of these terms and how their values were chosen?

Response 3: We thank the referee for this comment. Our methods section introduces D^T and D^R as the translational and rotational diffusion constant, respectively, associated with translational and rotational noise. In experiments, translational and rotational noise can arise from multiple sources in a quasi-2D vertically vibrated systems. Any asymmetric or off-axis particle–plate contacts during vertical vibration, where a spinner collides with the substrate at finite angles rather than perfectly normal incidence, can generate random forces and torques on the spinner. In addition, sub-micron level inhomogeneities in the substrate and spinner surfaces, which locally modify frictional contacts, may further contribute to translational and rotational noise in the system.

In our simulations, the diffusion coefficients D^T and D^R were selected within a parameter range that reproduces experimentally observed behavior, such as self-shearing and stable scar configurations, with the results remaining qualitatively unchanged over this range.

Comment 4 : Supplementary Fig. S1 shows the distribution of same-type near-

est neighbours at $\chi = 0$, $\phi = 0.72$. The authors should clarify whether these distributions were computed over the entire system or only the bulk region.

Response 4: We thank the referee for this comment. The distribution shown in Supplementary Fig. S1 was computed over the entire system. This has now been explicitly stated in the revised Supplementary Information.